

# Improving the Xin'anjiang Hydrological Model Based on Mass-Energy Balance

Yuan-Hao Fang[1,4], Xingnan Zhang[1,2,3], Chiara Corbari[4], Marco Mancini[4], Guo-Yue Niu[5,6], and Wenzhi Zeng[7]

[1]Department of Hydrology and Water Resources, Hohai University, Nanjing, China
[2]National Engineering Research Center of Water Resources Efficient Utilization and Engineering Safety, Hohai University, Nanjing, China
[3]National Cooperative Innovation Center for Water Safety & Hydro-Science, Hohai University, Nanjing, China
[4]Department of Civil and Environmental Engineering (D.I.C.A.), Politecnico di Milano, Milan, Italy
[5]Department of Hydrology and Water Resources, University of Arizona, Tucson, Arizona, USA
[6]Biosphere 2, University of Arizona, Oracle, Arizona, USA
[7]State Key Laboratory of Water Resources and Hydropower Engineering Science, Wuhan University, Wuhan, China

*Correspondence to:* Xingnan Zhang (zxn@hhu.edu.cn)
                    Chiara Corbari (chiara.corbari@polimi.it)

**Abstract.** Conceptual hydrological models are preferable for real-time flood forecasting, among which the Xin'anjiang (XAJ) model has been widely applied in humid and semi-humid regions of China. Although the relatively simple mass balance scheme ensures a good performance of runoff simulation during flood events, the model still has some defects. Previous studies have confirmed the importance of Evapotranspiration (ET) and soil moisture content (SMC) in runoff simulation. To overcome the

defects of the original XAJ model, an energy balance scheme suitable for the XAJ model was developed and coupled with the original mass balance scheme of the XAJ model. The detailed parameterizations of the improved model, XAJ-EB, are presented in the first part of this paper. XAJ-EB employs various meteorological forcing and remote sensing data as input, simulating ET and runoff yield using a more physically-based mass-energy balance scheme. In particular, the energy balance is solved by determining the representative equilibrium temperature (RET), which is comparable to land surface temperature (LST).

The XAJ-EB was evaluated in the Lushui catchment situated in the middle reach of the Yangtze River Basin for the period between 2004 and 2007. Validation using ground-measured runoff data proves that the XAJ-EB is capable of reproducing runoff comparable to the original XAJ model. Additionally, RET simulated by XAJ-EB agreed well with Moderate Resolution Imaging Spectroradiometer (MODIS) retrieved LST, which further confirms that the model is able to simulate the mass-energy balance since LST reflects the interactions among various processes. The validation results prove that the XAJ-EB model has

superior performance compared with the XAJ model and also extends its application fields.

## 1   Introduction

Hydrological models are widely used for real-time flood forecasting due to their abilities to predict hydrological fluxes (e.g. runoff) and states (e.g. soil moisture) with various leading time (Chen et al., 2016). These models can be grouped as physically-





based models which are mainly based on partial differential equations (e.g. Richards equation, de St. Venant equation, etc.) or conceptual models which usually employ a number of mathematical functions or distribution curves to reproduce hydrological processes (Kampf and Burges, 2007; Niu et al., 2014). Conceptual models are preferred for flood forecasting with consideration of the data and computational conditions (Blöschl et al., 2008; Hapuarachchi et al., 2011). Examples are the Sacramento soil

moisture accounting (SAC-SMA) model (Burnash et al., 1973) implemented by the US National Weather Service (NWS) (Smith et al., 2003); a spatially distributed flash flood model used in northern Austria (Blöschl et al., 2008); the HBV model (Bergström et al., 1995; ) adopted for the forecasting of Savinja catchment (Kobold and Brilly, 2006) and the Xin'anjiang (XAJ) model (Zhao, 1995) applied in the Huaihe river basin, China (Lu et al., 2008). These operational practices have proved the accuracy of stream discharge predicted by conceptual models, which is usually the major concern of real-time flood forecasting.

In China, the XAJ model is the most widely used model for flood forecasting in humid and semi-humid regions (Liu et al., 2009; Yao et al., 2009; Zhao, 1992). The XAJ model employs a spatial probability distribution curve to represent the variability of tension water capacity in the catchment and calculates runoff generation based on the saturation-excess mechanism (Zhao, 1995). With respect to the evapotranspiration (ET), the XAJ model uses pan evaporation as its input, and then computes actual ET using an empirical relationship, taking only the soil moisture content (SMC) into account. Such a generalized scheme

successfully strikes a balance between model complexity and computational accuracy, providing a reasonable runoff prediction during flood events after proper calibration.

At the catchment scale, although precipitation and runoff can be measured by traditional approaches, accurate catchment average ET and SMC are difficult to obtain. Therefore the bias of simulations is more likely to be accumulated in SMC due to lack of constrains. The significance of SMC to real-time flood forecasting has gradually been recognized. Studies showed

that the bias in predicting flood peaks is related to unrealistic antecedent SMC estimation (Huza et al., 2014), and therefore the performance of real-time flood forecasting can be improved by setting or assimilating initial SMC (Brocca et al., 2009; Berthet et al., 2009; Komma et al., 2008; Tramblay et al., 2010; Wanders et al., 2014). The accuracy of SMC estimation before flood events largely depends on ET estimation. In addition, considering the abilities to extend the leading time and quantify predictability, ensemble flood forecasting techniques are more attractive today (Cloke and Pappenberger, 2009), and

the estimation of SMC and ET is even more important in ensemble flood forecasting due to a longer leading time. However, the simple and empirical ET routine of the XAJ model is not able to guarantee the precision of ET simulation, especially during non-rainfall periods before flood events, leading to bias in SMC estimation which further influences the runoff prediction. The major defects of the ET routine in the XAJ model are: (1) the input pan evaporation is measured only at few specific locations, reflecting daily evaporation from open water, which means that the potential ET (PET) over a large area is assumed to be

the same. Such an assumption dose not always hold under heterogeneous meteorology or underlying surface conditions (Xu et al., 2006; Yuan et al., 2008); (2) calibration of $K_c$ (see Section 2.1 for details), a sensitive parameter of the XAJ model controlling water balance, is needed to convert pan evaporation to PET, which is impossible for ungauged catchments where observed runoff is unavailable; and (3) the empirical relationship linking PET with actual ET only takes water balance into account, neglecting other factors (e.g. meteorological conditions) that control ET processes (Wang and Dickinson, 2012). For

the aforementioned reasons, it is therefore necessary to improve the original ET routine of the XAJ model.



It should be noted that the physically-based ET schemes have been intensively studied in the land surface modeling community (Overgaard et al., 2006). Land surface models (LSMs) are developed to provide various fluxes and states connecting the atmosphere and land surface (Overgaard et al., 2006; Niu et al., 2011). Most LSMs have an energy balance component for ET estimation, but the way these models solve the energy balance differs. According to Su (2002) and Kalma et al. (2008),

generally three different approaches are employed by LSMs for ET estimation: (1) calculate all energy balance components except latent heat flux, which is obtained as the residual of the energy budget; (2) compute all components involved in energy balance by closing the balance equation, latent heat is solved at the same time when energy budget is closed; and (3) an empirical approach using water stress to derive ET. However, these approaches are rarely applied in hydrological models,especially for real-time flood forecasting, because their structures are complex and generally require considerable data and parameters to

drive the model.

Benefiting from remote sensing and data assimilation techniques, we now have more meteorological and land surface data. The scientific community has been working to improve ET scheme of hydrological models (e.g. Corbari et al., 2011; Niu et al., 2014; Spies et al., 2015; Yan et al., 2012). In particular, some efforts have been made to improve the ET simulation of the XAJ model. Methodologies reported can be summarized as two approaches. The first approach was to introduce a physically-based

formula to simulate PET based on meteorological measurements, aiming to provide more accurate PET input while the XAJ model structure remained unchanged (Yuan et al., 2008). The second approach involved replacing the ET routine by a more sophisticated scheme, typically the Penman-Monteith (PM) equation, which simulates actual ET by meteorological variables, remote sensing data and modeled SMC (Li et al., 2009; Zhou et al., 2013). These studies have demonstrated the feasibility of improving the ET scheme of the XAJ model, and the second approach, which replaces the overall ET routine, is superior

because the actual ET process is better parameterized. However, the PM equation tends to neglect evaporation due to the "big leaf" assumption (Yan et al., 2012). In addition, given that the actual ET process is controlled by both water and energy, a more scientific way to simulate ET is by using both the mass and energy balance. Such problems of PM equation can be further overcome by introducing an energy balance-based scheme. For example, as reported by Corbari et al. (2011), a water balance model FEST (Rabuffetti et al., 2008) was augmented by coupling a energy balance scheme and various case studies

have confirmed its applicability under different conditions (Masseroni et al., 2011; Corbari et al., 2013; Corbari and Mancini, 2014b).

In this paper, we developed an energy balance scheme suitable for the XAJ model and coupled this to runoff yield scheme of the original model, replacing the previous empirical ET scheme. The improved model employs a physically-based mass-energy balance component to simulate ET and runoff yield. The remainder of the paper is organized as follows: Section 2 presents

the basic theory we adopted to develop the mass-energy balance scheme for the XAJ model; Section 3 reports the calibration and validation of the improved model against various observations; Section 4 further discusses the advantages of the improved model and Section 5 summarizes the study.



## 2  Improving the XAJ model

### 2.1   the XAJ Model and Its Mass Balance Scheme

The XAJ model was developed by Zhao (1977, 1995) based on the concept of runoff formation with respect to repletion of storage, which means that for each location in the catchment, there is no runoff yielded until the soil water deficit is replenished.

Therefore, the XAJ model is the most suitable for humid and semi-humid regions where saturation-excess runoff is more likely to occur. A statistical tension water capacity curve was introduced to represent the spatial distribution of tension water capacity (maximum soil water deficit, i.e. the difference between field capacity and wilting point), which is regarded as the essence of the XAJ model. The flow chart of the XAJ model is shown in Figure 1. All symbols outside the blocks are parameters, whose physical meanings are shown in Table 1. The inputs to the model are areal mean rainfall (P) and measured pan evaporation

(EM) while the outputs are the discharge at the outlet of the basin (TQ) and the actual ET.

**Figure 1 here**

**Table 1 here**

The basic computational unit of the XAJ model is the element area, which, in principal, is a small natural catchment that has relatively homogeneous underlying surface characteristics (e.g. terrain, soil and vegetation, etc.). Simulation of outflow from

each element area consists of 4 major components (Li et al., 2011; Qu et al., 2011). Here we only present a simple description; for mote details, refer to Zhao (1992):

1. Evapotranspiration, which is simulated by a three-layer soil (i.e. upper, lower, and deep layer) model based on pan evaporation and soil moisture;

2. Runoff yield, which, based on the tension water capacity curve, simulates the runoff yield according to the rainfall and

soil storage deficit;

3. Runoff separation, which separates the abovementioned runoff into three components, i.e., surface, subsurface, and groundwater;

4. Flow routing, which transfers the local runoff to the outlet of each basin forming the outflow. Several approaches including a unit hydrograph, linear reservoir, and lag & route can be adopted.

The mass balance of the XAJ model is expressed as:

$$\Delta W = P - R - ET \tag{1}$$

where $\Delta W$ is the soil water content storage term (mm); $P$ is precipitation (mm); and $R$ is runoff yield (mm).





The mass balance solution depends on the non-linear relationship between $W$ and $R$ represented by a tension water capacity curve (Figure 2), for a given time step when $P$ is larger than $ET$, $R$ is calculated as:

$$R = P - ET - (WM - W_0) + WM\left(1 - \frac{P - ET + A}{WMM}\right)^{1+B} \qquad (P - ET + A < WMM) \qquad (2)$$

$$R = P - ET - (WM - W_0) \qquad (P - ET + A \geqslant WMM) \qquad (3)$$

where $W_0$ is the initial soil water (mm), $A$ is the value of Y axis of the tension water capacity curve corresponding to $W_0$ (mm), $WMM$ is the maximum tension water capacity over the catchment (mm); and $WM$ and $B$ are parameters of the XAJ model as listed in Table 1.

**Figure 2 here**

## 2.2 Energy Balance Scheme Developed for the XAJ model

The energy balance of land surface is expressed as:

$$\Delta S = R_n - G - H - LE \qquad (4)$$

where $\Delta S$ is the energy storage term (W m$^{-2}$); $R_n$ is net radiation (W m$^{-2}$); $G$ is ground heat flux (W m$^{-2}$); $H$ is sensible heat flux (W m$^{-2}$); and $LE$ is latent heat flux (W m$^{-2}$).

In this paper, we adopted a "patch approach" to distinguish the energy fluxes between bare soil and the canopy (Lhomme

and Chehbouni, 1999; Lu et al., 2014), assuming both bare soil and the canopy receive the same radiation loading, and the total sensible and latent heat fluxes are weighted by the canopy fraction $f_v$ derived from leaf area index (LAI):

$$f_v = 1 - e^{-0.52LAI} \qquad (5)$$

1. **Net radiation** ($R_n$)

    $R_n$ is the arithmetic difference between downward and upward short and long wave radiation:

$$R_n = R_{ds}(1 - \alpha) + R_{dl} - \zeta\sigma(RET^4) \qquad (6)$$

    where $R_{ds}$ and $R_{dl}$ are downward short and long wave radiation (W m$^{-2}$), respectively; $\alpha$ is land surface Albedo (-); $\zeta$ is land surface emissivity (-); $\sigma$ is the Stefan-Boltzmann constant (W m$^{-2}$ K$^{-4}$); and RET is representative equilibrium temperature (K).

2. **Ground heat flux** ($G$)

$G$ is the flux that is transferred between the land surface and the subsurface via soil thermal conduction:

    $$G = (k_s/dz)(RET - T_{soil}) \qquad (7)$$





Where $k_s$ is soil thermal conductivity (W m$^{-1}$ K$^{-1}$), which is related to soil conditions (Corbari et al., 2011; McCumber and Pielke, 1981); $dz$ is the soil depth for calculating ground heat flux (m); and $T_{soil}$ is soil temperature at depth $dz$ (K).

Eq.(7) can be solved numerically together with the heat diffusion equation as implemented by many LSMs; however, such an approach requires detailed thermal and hydrological information on different soil layers which is not available in the XAJ model. In addition to the numerical solution, there are also other parameterizations that derive $G$ from more easily available data, e.g. net radiation (Idso et al., 1975; Santanello and Friedl, 2003; Su, 2002), sensible heat flux (Cellier et al., 1996) or surface temperature (Bhumralkar, 1975; Deardorff, 1978; Wang and Bras, 1999). Liebethal and Foken (2007) and Venegas et al. (2012) evaluated different approaches and found these alternatives can also reproduce reasonable $G$ after calibration. In order to accommodate the energy balance scheme, we adopted force restore model proposed by Bhumralkar (1975) and Blackadar and Blackadar (1976) to estimate $G$ from $RET$, the original force restore equation to estimate soil temperature can be rearranged as:

$$G = \frac{1}{C_T}\left[\frac{(RET - RET_0)}{\Delta t} + \frac{2\pi}{\tau}(RET - \overline{T})\right] \tag{8}$$

where $RET_0$ is the representative equilibrium temperature of the previous time step (K); $\Delta t$ is time step (s); $\frac{2\pi}{\tau}$ is the angular frequency for diurnal forcing (radians s$^{-1}$); $\overline{T}$ is the mean surface temperature (K); and $C_T$ is the coefficient weighted by the volumetric heat capacity of soil and canopy:

$$C_T = 1/\left(\frac{1 - f_v}{C_g} + \frac{f_v}{C_v}\right) \tag{9}$$

where $C_g$ is the soil heat capacity (MJ m$^{-3}$ K$^{-1}$) and $C_v$ is the canopy heat capacity (MJ m$^{-3}$ K$^{-1}$).

3. **Sensible heat flux ($H$)**

$H$ represents heat energy transferred between the surface and air when their temperatures are different, which is the weighted average of the sensible heat flux of bare soil and the canopy:

$$H = (1 - f_v)H_s + f_v H_c \tag{10}$$

where $H_s$ and $H_c$ are the sensible heat flux of bare soil and the canopy (W m$^{-2}$), respectively, which are parameterized as:

$$H_s = \frac{\rho_a c_p}{r_{abs}}(RET - T_a) \tag{11}$$

$$H_c = \frac{\rho_a c_p}{r_a}(RET - T_a) \tag{12}$$

where $\rho_a$ is air density (kg m$^{-3}$); $c_p$ is the specific heat capacity of air (MJ kg$^{-1}$ K$^{-1}$); $T_a$ is air temperature (K); $r_{abs}$ and $r_a$ are the aerodynamic resistances for bare soil and canopy (s m$^{-1}$), respectively.

The aerodynamic resistance determines the transfer of heat and water vapor from evapotranspiration surface into the air at reference height. For the canopy component, $r_a$ is evaluated according to Thom (1972) as:

$$r_a = \frac{[\ln(\frac{z_m - d}{z_{om}}) - \Psi_m(\frac{z_m - d}{L})][\ln(\frac{z_h - d}{z_{oh}}) - \Psi_h(\frac{z_h - d}{L})]}{k^2 u} \tag{13}$$





where $z_m$ and $z_h$ are the reference heights where wind and humidity are measured (m); $d$ is the zero plane displacement height (m); $z_{om}$ and $z_{oh}$ are the roughness length governing the transfer of momentum and heat, respectively (m); $\Psi_m$ and $\Psi_h$ are atmospheric stability correction factors for momentum and heat (-), respectively; $L$ is the Obukhov length (m); $k$ is the Von Karman constant; and $u$ is wind speed (m s$^{-1}$).

In this paper, we estimated $d$, $z_{om}$ and $z_{oh}$ by empirical functions based on canopy height h (m) (Allen et al., 1998) :

$$d = 0.666h \tag{14}$$

$$z_{om} = 0.123h \tag{15}$$

$$z_{oh} = 0.1z_{om} \tag{16}$$

The aerodynamic resistance for bare soil $r_{abs}$ can be determined in the same way as $r_a$ using Eq. (13), but with different roughness length for bare soil. In this paper, we assumed 0.01 m and 0.001 m for $z_{om}$ and $z_{oh}$, respectively.

4. **Latent heat flux** ($LE$)

$LE$ is the energy used for the phase change of water, which is directly related to ET. It is also the weighted average of the latent heat flux of bare soil and the canopy:

$$LE = (1 - f_v)LE_s + f_vLE_c \tag{17}$$

where $LE_s$ and $LE_c$ are the latent heat fluxes of bare soil and the canopy (W m$^{-2}$), respectively, which are parameterized following Corbari et al. (2011) as:

$$LE_s = \frac{\rho_a c_p}{\gamma(r_{abs} + r_s)}(e^* - e_a) \tag{18}$$

$$LE_c = \frac{\rho_a c_p}{\gamma(r_a + r_c)}(e^* - e_a) \tag{19}$$

where $\gamma$ is the psychometric constant (Pa °C$^{-1}$); $r_s$ and $r_c$ are resistances for bare soil and the canopy (s m$^{-1}$), respectively; $e^*$ is the saturated vapor pressure of the evapotranspiration surface (Pa); and $e_a$ is the vapor pressure of air (Pa). In particular, $e^*$ is also related to RET:

$$e^* = 6.11 \times 10^{\frac{7.5RET_c}{237.3+RET_c}} \tag{20}$$

where $RET_c$ is RET expressed in degree Celsius (°C)

In this paper, we parameterized $r_s$ and $r_c$ following the work of Corbari et al. (2011) and Yan et al. (2012), owing to the similar soil water routines of the XAJ model and models reported by these authors.

$$r_s = 3.5 \left(\frac{\theta_{sat}}{\theta}\right)^{2.3} + 33.5 \tag{21}$$

$$r_c = \frac{1}{R_h} \frac{r_{s\ min}}{LAI} \frac{\theta_{fc} - \theta_{wp}}{\theta - \theta_{wp}} \tag{22}$$





where $\theta_s$ is saturated soil moisture (-); $\theta$ is soil moisture (-); $r_{s\ min}$ is minimum stomatal resistance of canopy (s m$^{-1}$); $R_h$ is relative humidity (-); and $\theta_{fc}$ and $\theta_{wp}$ are field capacity and wilting point (-), respectively.

As we summarized in Section 1, there are several approaches to derive latent heat flux from the energy balance. In this paper, we adopted the approach proposed by Corbari et al. (2011) where the energy balance is solved by determining RET, which is theoretically the LST that closes the balance. Given that the energy storage term ($\Delta S$) is often negligible at the basin scale (Corbari et al., 2011) and the remaining energy budget components in Eq. (4) are all related to RET, we employed Newton-Raphson iterative method to solve RET that can close the energy balance. As reported by Corbari et al. (2011), the Newton-Raphson method is an efficient way to solve the energy balance under different hydro-meteorological conditions. The actual LE is then solved based on the resulting RET using Eq. (17) through Eq. (22).

### 2.2.1 XAJ-EB: the XAJ Model Improved by Coupling Mass-Energy Balance

The mass (Eq. 1) and energy balance (Eq. 4) is coupled through $ET$ and $W$. $ET$ is derived from $LE$ in the energy balance as:

$$ET = \frac{LE}{\lambda \rho_w} \tag{23}$$

where $\lambda$ is the latent heat of vaporization (MJ kg$^{-1}$); and $\rho_w$ is the density of water (kg m$^{-3}$).

For a given time step, based on W of the previous time step, the energy balance calculates ET and transfers it to the mass balance to update W. The updated W is transferred back to the energy balance to calculate ET until the coupled mass-energy balance is achieved, i.e. difference in W from the last two iterations is below a pre-defined threshold (0.01 in this paper).

In order to couple the energy balance we developed to the XAJ model, we changed the basic computational unit for runoff yield, from an elementary area to a grid cell. The computational unit of the energy balance scheme is grid because all inputs to it are grid-based, which is different from that of the XAJ model. We adopted the "Grid-XAJ" concept that also employs the grid as a computational unit (Li et al., 2007; Yao et al., 2009). Here, we used a grid to compute runoff yield only, rather than all processes, because runoff separation and routing are isolated from runoff yield and do not affect the mass balance of the grid.

The improved XAJ model, i.e. XAJ-EB introduces atmospheric forcing and remote sensing data as input and calculates runoff, evapotranspiration and soil water simultaneously using a grid cell based on the mass-energy balance. Runoff yield calculated for grid cells are aggregated to an elementary area for routing simulation.

By coupling the energy balance scheme, XAJ-EB is able to calculate ET based on meteorological and remote sensing data, providing ET estimation at high spatial and temporal resolution, which successfully overcomes the defects of original ET scheme of the XAJ model. More importantly, as a key variable of the energy balance solution, RET represents the equilibrium temperature of the land surface that controls the entire mass-energy balance. RET is comparable to land surface temperature (LST) retrieved from remotely-sensed imagines (Corbari et al., 2011), which serves as a new constraint of the XAJ model besides runoff.





## 3 Evaluation of the XAJ-EB Model

### 3.1 Study Area

We selected a gauged catchment, namely the LuShui river catchment (LS) to test the XAJ-EB model. It is situated in the the middle reaches of Yangtze River (Chang Jiang) Basin and is controlled by the ChongYang hydrological site (Figure 3). LS covers an area of 2250 km$^2$, ranging from 29.08N to 29.83N and 113.67E to 114.17E. Annual precipitation, pan evaporation and runoff depth of LS are 1550 mm, 1200 mm and 753.3 mm, respectively (Cheng et al., 2013). The study area is mainly characterized by mountain and hill terrain which covers more than $90\%$ the total area, with a mean elevation of 258 m. According to MODIS-based land cover climatology data (Broxton et al., 2014), the major land cover types of this area are crop land and mixed forests. There are 8 precipitation sites and 1 hydrological site within the catchment, operated by the Bureau of Hydrology, Yangtze River Water Resources Commission.

**Figure 3 here**

### 3.2 Data

#### 3.2.1 Digital Elevation Model (DEM)

We mainly relied on digital elevation model (DEM) to delineate the sub-catchments and elementary areas (the basic computational unit of the XAJ model, see Section 2.1 for details). The DEM dataset we employed in this paper was ASTER (Advanced Spaceborne Thermal Emission and Reflection Radiometer) DEM Version 2, released in October, 2011 by the U.S. National Aeronautics and Space Administration (NASA) and Japan's Ministry of Economy, Trade, and Industry (METI). The spatial resolution of ASTER DEM is 1″ (approximately 30 m in the study area). DEM data were downloaded from http://dx.doi.org/10.5067/ASTER/ASTGTM.002 and extracted based on the boundary of LS. To match the spatial scale of the model (see Section 3.3 for details), we further resampled the DEM to the spatial resolution of 1 km using majority resampling technique.

#### 3.2.2 Land Cover and Soil Data

Land cover and soil data were used to determine several land cover or soil-dependent parameters (e.g. minimal stomatal resistance and soil tension water capacity) of the model.

The land cover dataset we chose was 0.5 km MODIS-based Global Land Cover Climatology developed by Broxton et al. (2014) based on 10 years (2001-2010) of the MODIS land cover (MCD12Q1, Collection 5.1) product. The dataset classified the land surface into 17 different land cover types, avoiding the unrealistic land cover change as observed by the original MCD12Q1 product. The land cover dataset is available online at http://landcover.usgs.gov/global_climatology.php. Similar to the DEM data, we resampled the land cover data into 1 km spatial resolution using majority resampling technique.

We mainly used 2 types of soil data properties for this study, i.e. soil physical properties (e.g. field capacity) which were obtained from a data set developed by Dai et al. (2013), and soil thickness obtained from a data set developed by Pelletier



et al. (2015). The former data set, namely the "China data set of soil properties", was developed mainly for land surface modeling and includes various soil hydraulic parameters derived from soil physical and chemical properties using pedotransfer functions (Dai et al., 2013). The spatial resolution of this dataset is 30″ (approximately 1 km in the study area) and the vertical variation of the soil properties is documented for 7 layers to a depth of 1.38 m. The soil properties data were retrieved

from http://globalchange.bnu.edu.cn/research/soil3. The latter data set, namely the "gridded global data set of soil, immobile regolith, and sedimentary deposit thicknesses", documents the estimation of the thickness of the permeable layers above the bedrock (Pelletier et al., 2015), which can be regarded as soil depth defined by the XAJ model. This dataset was retrieved from http://dx.doi.org/10.3334/ORNLDAAC/1304, and it has the same spatial resolution as the China data set of soil properties at 30″.

### 3.2.3   Remote Sensing Data

Variables retrieved from remote sensing data were used to drive (e.g. leaf area index (LAI) and Albedo) or validate the model (e.g. land surface temperature (LST)). We adopted various Moderate Resolution Imaging Spectroradiometer (MODIS) products to provide spatial estimations of LAI, Albedo and LST; detailed information on these variables, including product name and spatial and temporal resolution can be found in Table 2. All remote sensing data were download from http://reverb.echo.nasa.

gov/ for the period between year 2004 and 2007. The Albedo dataset was resampled to a 1 km spatial resolution.

**Table 2 here**

### 3.2.4   Meteorological and Hydrological Data

The meteorological forcing data we employed included precipitation, downward short wave radiation, downward long wave radiation, wind speed, air temperature, air pressure and specific humidity from 2004 to 2007. Precipitation data from 8 rain

gauges were collected by the Bureau of Hydrology, Yangtze River Water Resources Commission. All other forcing data were retrieved from the China Meteorological Forcing Dataset (CMFD) (He and Yang, 2011) which was produced by merging a variety of data sources (Chen et al., 2011; Leng et al., 2015).The spatial and temporal resolutions of this dataset are 0.1° (approximately 10 km in the study area) and 3 hours, respectively. The dataset was downloaded from http://westdc.westgis.ac. cn/. Runoff and pan evaporation data of the ChongYang hydrological station for the same period were also collected by the

Bureau of Hydrology.

### 3.3   Model Setup

In consideration of the catchment characteristics as well as data availability, we defined the dimension of the computational grid as 1 km × 1 km, resulting in 72 columns × 60 rows that covered the study area. All other datasets were resampled to 1 km × 1 km resolution. The temporal resolution of the model is 3 h, the same as the meteorological forcing data we

employed.





As seen from Eq. 4 through Eq. 22, a range of parameters/variables is needed for the energy balance scheme. Each grid was assigned a set of time-independent parameters including soil physical properties (e.g. $\theta_{fc}$), vegetation properties (e.g. $r_{s\,min}$) based on soil and vegetation type. Other time-dependent variables were obtained either from remote sensing data (e.g. LAI) or model simulated states (e.g. $\theta$).

## 3.4 Calibration and Validation of the XAJ Model

As listed in Table 1, several parameters have to be estimated before applying the XAJ model, among which tension water capacity $WM$ has physical definition that can be estimated from the soil proprieties for each grid:

$$WM = (\theta_{fc} - \theta_{wp}) \times SD \tag{24}$$

where SD is soil depth (mm). All three soil proprieties can be retrieved from soil dataset we described above.

For parameters other than $WM$, we chose two years of data (2004 and 2005) to perform a calibration against observed runoff data using the original XAJ model, which calculates ET using measured pan evaporation. We first introduced a model-independent parameter estimation tool, namely PEST (Doherty et al., 1994) to provide an optimized combination of parameters. PEST is based on the Gauss-Marquardt Levenberg (GML) algorithm (Marquardt, 1963) and has been widely applied in calibrating hydrological models. The initial values as well as the optimization limits of the parameters were set according to Zhao (1984). After the automatic calibration, we used the traditional trial and error method to adjust some parameters based on our experience in calibrating the XAJ model. The metrics we adopted to evaluate the model performance are the root-mean-square error (RMSE), Nash-Sutcliffe model efficiency coefficient (NSE) and relative error of total runoff volume (bias):

$$RMSE = \sqrt{\frac{1}{n}\sum_{i=1}^{n}(Q_{obs,i} - Q_{sim,i})^2} \tag{25}$$

$$NSE = 1 - \frac{\sum\limits_{i=1}^{n}(Q_{obs,i} - Q_{sim,i})^2}{\sum\limits_{i=1}^{n}(Q_{obs,i} - \overline{Q}_{obs})^2} \tag{26}$$

$$bias = \frac{V_{sim} - V_{obs}}{V_{obs}} \tag{27}$$

$$V_{sim} = \sum_{i=1}^{n-1}\left(\frac{Q_{sim,1} + Q_{sim,i+1}}{2}\right)\Delta t_i \tag{28}$$

$$V_{obs} = \sum_{i=1}^{n-1}\left(\frac{Q_{obs,1} + Q_{obs,i+1}}{2}\right)\Delta t_i \tag{29}$$

where $Q_{obs,i}$ and $Q_{sim,i}$ are observed and modeled discharge (m$^3$ s$^{-1}$) at time step $i$, respectively; $n$ is total time step; and $\Delta t_i$ is the interval between time step $i$ and $i+1$ (s).

Table 1 summaries the optimized parameter values while Figure 4a shows comparison between XAJ-modeled and observed runoff during the calibration period. The XAJ model reproduced both the variation and amplitude of flood events with RMSE,




NSE and bias values of 37.96 m$^3$ s$^{-1}$, 0.70 and $-0.09$ %, respectively, which indicates the good performance of the model and the efficiency of our calibration strategy. One important objective of the calibration process was to control the overall water balance (-0.09%) by adjusting $KC$ based on pan evaporation, because the model performance is sensitive to the accuracy and representativeness of the pan evaporation observations.

5    To validate the model, we ran the model for another two years (2006 and 2007) with these calibrated parameters. Figure 4b presents the validation results which shows the good agreement between XAJ-modeled and observed runoff, although the XAJ model slightly overestimated the total runoff volume by 2.18%, the RMSE and NSE were even better than the values obtained during the calibration period, further confirming the fitness and robustness of the parameters we calibrated.

**Figure 4 here**

10  **3.5   Validation of XAJ-EB Against Runoff**

Given that the mass balance of the XAJ model remained unchanged, we used the calibrated parameters directly to run the XAJ-EB model for the whole period between 2004 and 2007. For this simulation using XAJ-EB, the coefficient $KC$ was eliminated since the ET was simulated directly using the mass-energy balance scheme. Figure 5 shows the comparison between XAJ-EB modeled and observed runoff. The overall RMSE, NSE and bias ewre 26.09 m$^3$ s$^{-1}$, 0.77 and $-0.53\%$, respectively. The 15  overall performance of the runoff simulation by XAJ-EB is comparable to that of the original XAJ model.

**Figure 5 here**

**3.6   Validation of XAJ-EB Against MODIS LST**

The RET simulated by XAJ-EB is theoretically the LST that closes the energy balance, which is comparable to the LST retrieved from remote-sensing data, providing another variable for model calibration and validation. Among various LST 20  products released, the MODIS LST products have been widely used owing to high accuracy (Corbari et al., 2011, 2014b; Wan et al., 2004). The dataset we used in this paper is the MOD11A1 daytime LST product. Although it is available daily at a 1 km spatial resolution, some images are affected by cloud, resulting in a high number of missing values. In order to better validate the LST simulation, we examined each image for the whole simulation period and chose 107 images with a maximum of 30% missing values.

25    We first performed grid-by-grid comparison using the sampling XAJ-EB modeled LST according to MODIS LST data availability, the fitness was evaluated by coefficient of determination ($R^2$):

$$R^2 = \left[\frac{n(\sum\limits_{i=1}^{n} LST_{sim,i}LST_{obs,i}) - \sum\limits_{i=1}^{n} LST_{sim,i} \sum\limits_{i=1}^{n} LST_{obs,i}}{\sqrt{[n(\sum\limits_{i=1}^{n} LST_{sim,i}^2) - (\sum\limits_{i=1}^{n} LST_{sim,i})^2][n(\sum\limits_{i=1}^{n} LST_{obs,i}^2) - (\sum\limits_{i=1}^{n} LST_{obs,i})^2]}}\right]^2 \tag{30}$$

where $LST_{sim_i}$ and $LST_{obs_i}$ are XAJ-EB-modeled and MODI-retrieved LST (K), respectively at time step i; and n is the total LST data to be evaluated.



Figure 6a shows the scatter plot between the XAJ-EB modeled and MODIS retrieved grid LST for all 107 MODIS images we chose; the resulting $R^2$ value reached reaches 0.91, indicating the good agreement between the two variables. We also plotlted the catchment average LST (Figure 6b), from which we can clearly see that most LST points are distributed along the 1:1 line with small deviation. Figure 6 indicates that XAJ-EB is capable of accurately simulating LST under various

hydrometeorological and underlying surface conditions.

We also plotted the time series of XAJ-EB-modeled catchment-average LST against MODIS-retrieved catchment-average LST (Figure 7), and calculated the corresponding RMSE and NSE values as 2.25 K and 0.89, respectively. The results are acceptable since the overall accuracy of the MODIS LST product is reported to be $\pm 1$ K (Kalma et al., 2008), which further confirm the fit between XAJ-EB modeled and MODIS retrieved LST.

**Figure 6 here**

**Figure 7 here**

### 3.7   Validation of XAJ-EB Against MODIS ET

There were no direct ET measurements in the LS catchment due to the lack of eddy flux towers. Consequently, to evaluate the ET simulation from XAJ-EB, we adopted the MODIS ET(MOD16A2) product as the reference. Different from the MODIS

LST product, MODIS ET is based on the P-M equation using various MODIS products(Mu et al., 2007, 2011). Although many studies have confirmed the overall accuracy of the product, the specific accuracy over some regions cannot always be guaranteed due to the algorithm itself as well as land surface characteristics (Corbari et al., 2014b; Mu et al., 2007, 2011; Ramoelo et al., 2014; Roux et al., 2013).

Figure 8 shows the catchment average of XAJ-EB modeled and MODIS-estimated 8-day ET, we also included the XAJ-

modeled ET here for comparison. Total MODIS-estimated ET for this period is 3234.8 mm, higher than the 2655.7 mm from XAJ-EB and 2393.4 mm from XAJ. However, the cumulative observed precipitation and runoff values for the whole study period were 4609.0 mm and 2193.3 mm, respectively, from which we can roughly estimate the cumulative ET for the same period as 2416.7 mm if we assume total precipitation can be balanced by ET and runoff over such a long period. Corbari et al. (2014b) reported an overestimate of MODIS ET for the Yangtze River basin, which is in accordance with our results

as shown in Figure 8, i.e. MODIS ET is higher than ET from both models. Although there is bias in the total ET estimation, XAJ-EB-modeled ET had an $R^2$ value of 0.70, higher than that of the XAJ modeled ET (0.50), which means the variation in ET from XAJ-EB was close to MODIS-estimated ET.

**Figure 8 here**





## 4    Discussion

### 4.1    Advantages of XAJ-EB over XAJ

In this paper, the original XAJ model was improved by coupling the mass-energy balance scheme. Validations using both ground measured runoff and remotely-sensed LST have proved the performance of XAJ-EB. Compared with the original XAJ

model, there are some obvious advantages of the XAJ-EB model that overcome the several defects as we reported in Section 1. First the model is capable of providing more reliable ET at high spatial and temporal resolution based on meteorological forcing and remotely sensed data, which may influence the soil moisture and further influence the runoff simulation. During 2007, we can see from Figure 9b that the most significant difference occurred during the dry season when there was little precipitation, and the low SMC from XAJ-EB before the two largest flood events reduced the flood peak, which was closer to

the observations (Figure 9a).

**Figure 9 here**

Another advantage is that XAJ-EB is more suitable for use in ungauged basins, where either measured pan evaporation or runoff data are unavailable. This is because XAJ-EB simulates ET based on meteorological forgings and remotely sensed data, rather than measured pan evaporation used by XAJ. Moreover, LST is a crucial parameter controlling the mass-energy

balance, which reflects the interactions among different processes (Wang et al., 2009), Moreover, LST is also an indicator of SMC variation (Corbari et al., 2011; Sandholt et al., 2002). Therefore, the model-simulated LST provides an alternative way to calibrate and validate the model, especially when observed runoff data are unavailable. In fact, some efforts have been made to exploit the LST for hydrological modeling (Corbari and Mancini, 2014b; Corbari et al., 2014a; Silvestro et al., 2013; Wang et al., 2009), and these studies have demonstrated the possibility of using LST as a supplement to traditional runoff data.

Finally, the XAJ model is not for flood foresting only, it has been used for investigating the effects of climate (Peng and Xu, 2010) or land cover (Tian et al., 2013; Qu et al., 2011) change on stream flow; identifying the drought events (Duan and Mei, 2014) and examining the variability of SMC memory for wet and dry basins (Rahman et al., 2015). Such studies can benefit from reliable ET and SMC simulation. By explicitly take into consideration meteorological forcing, land cover and vegetation characteristics, XAJ-EB is more suitable than XAJ for the study of hydrological responses under changing climate/land cover,

which may help to extend the applications of original XAJ model.

### 4.2    Applicability of simplified energy balance of XAJ-EB

As we mentioned in Section 1, the energy balance scheme we developed for the XAJ model is physically-based with proper structures that are suitable for real-time flood forecasting operations. Consequently, we used certain generalizations and simplifications, especially for LST, we adopted a lumped RET to represent the integrated LST for the land surface. In contrast,

some LSMs involve more sophisticated schemes, e.g. Noah-MP LSM (Niu et al., 2011) introduced 3 different LSTs: $T_{g,b}$ for temperature of bare ground fraction, $T_{g,v}$ for temperature of the vegetated fraction and $T_v$ for canopy surface temperature.





To further investigate the applicability of XAJ-EB, we ran Noah-MP with the same dataset we used for XAJ-EB but excluded precipitation. Because Noah-MP requires grid precipitation input, we used the CMFD precipitation field rather than the gauge-measured value. Figure 10 presents the comparison of daily LST and latent heat flux from XAJ-EB with those from Noah-MP.

LST from Noah-MP was estimated using gridded upward long wave radiation which also represents the integrated LST
for the land surface. Figure 10a shows a generally good agreement between the two LSTs, with RMSE, NSE and bias values as 1.68 K, 0.97 and -0.20%, which indicates that compared with a more sophisticated LSM, the energy balance scheme of XAJ-EB is able to produce reliable LST with only one lumped temperature, which may also justify our generalizations and simplifications of the energy balance scheme.

As for the latent heat flux, although the overall bias was small (-2.53 %), low NSE (0.53) indicates there is an inconsistency
in inconsistence of the ET time series. This is partly due to the different precipitation fields we used for the 2 models, which had an NSE of only 0.51 (Figure 10c). By comparing Figure 10b with Figure 10c we found that a larger bias of ET generally corresponds to a larger bias in precipitation.

**Figure 10 here**

In addition to the simplification of the energy balance scheme, XAJ-EB also makes use of various remote sensing products
(i.e. LAI, Albedo, etc.) to eliminate the processes that have little effects on flood forecasting (e.g. vegetation dynamics), only retaining the essential processes that related to ET and runoff simulation, which help to reduce both the complexity of the model and number of parameters need calibrated.

## 5   Conclusion

In this paper, an energy balance based scheme suitable for the XAJ model was developed by explicitly taking account of
bare soil and the canopy using a "patch approach". Different energy fluxes for bare soil and the canopy respectively were parameterized. The energy balance was solved by determining RET, which is theoretically the LST that closes the balance. The energy balance scheme was then coupled to the mass mass balance scheme of the XAJ model. The improved model, XAJ-EB estimates the actual ET and runoff yield through a mass-energy balance approach using various meteorological and remotely sensed data.

Taking the LS catchment as the study area, we calibrated and validated the original XAJ model and used the same optimized parameters to evaluate the performance of the XAJ-EB model. With respect to the runoff simulation for the period between 2004 and 2007, the XAJ-EB model had RMSE, NSE and bias values of 26.09 $m^3$ $s^{-1}$, 0.77 and -0.53%, respectively. The results matched with the observed runoff well, which was also comparable to the original XAJ model. In addition to runoff, XAJ-EB is also capable of simulating the dynamics of LST with $R^2$, RMSE and NSE values of 0.93, 2.25 K and 0.89,
respectively compared with MODIS-retrieved catchment average LST. The good match between modeled and remotely-sensed LST implies that the XAJ-EB model is able to reproduce the mass-energy balance processes since LST reflects the interactions among various processes. Moreover, with LST as an output, XAJ-EB adds a new constraint and offers a potentially new approach for model calibration and validation, especially when runoff data are unavailable.





The mass-energy balance scheme developed in this paper is comparable to the sophisticated LSM Noah-MP model in terms of LST and latent heat flux modeling, which overcomes several defects of the original XAJ model that simulates actual ET using pan evaporation measurements. The inter-comparison between XAJ-EB and XAJ shows that the improvement of ET estimation can help to improve the runoff simulation, especially the runoff peak which is the major concern of real-time flood

forecasting. Moreover, by explicitly take consideration of different atmospheric and underlying surface conditions, XAJ-EB is more suitable than XAJ for the study of hydrological responses under changing climate/land cover, which may help to extend the applications of the original XAJ model.

*Author contributions.* Zhang and Corbari designed and supervised the study; Fang, Cobari and Mancini developed the model and conducted simulations; Zeng collected and processed the remote sensing and ground measured data; all authors contributed to the preparation of this

manuscript.

*Acknowledgements.* This study is funded by the Major International (Regional) Joint Research Project of the National Natural Science Foundation of China (51420105014), the Special Scientific Research Fund of Ministry of Water Resources' Public Welfare Profession of China (201401034), ESA-MOST Dragon 3 programme (10664), the National Natural Science Foundation of China (51609175), the Open Foundation of State Key Laboratory of Hydrology-Water Resources and Hydraulic Engineering (2015490211), and the Research and

15 Innovation Project for College Graduates of Jiangsu Province(2014B35114).



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



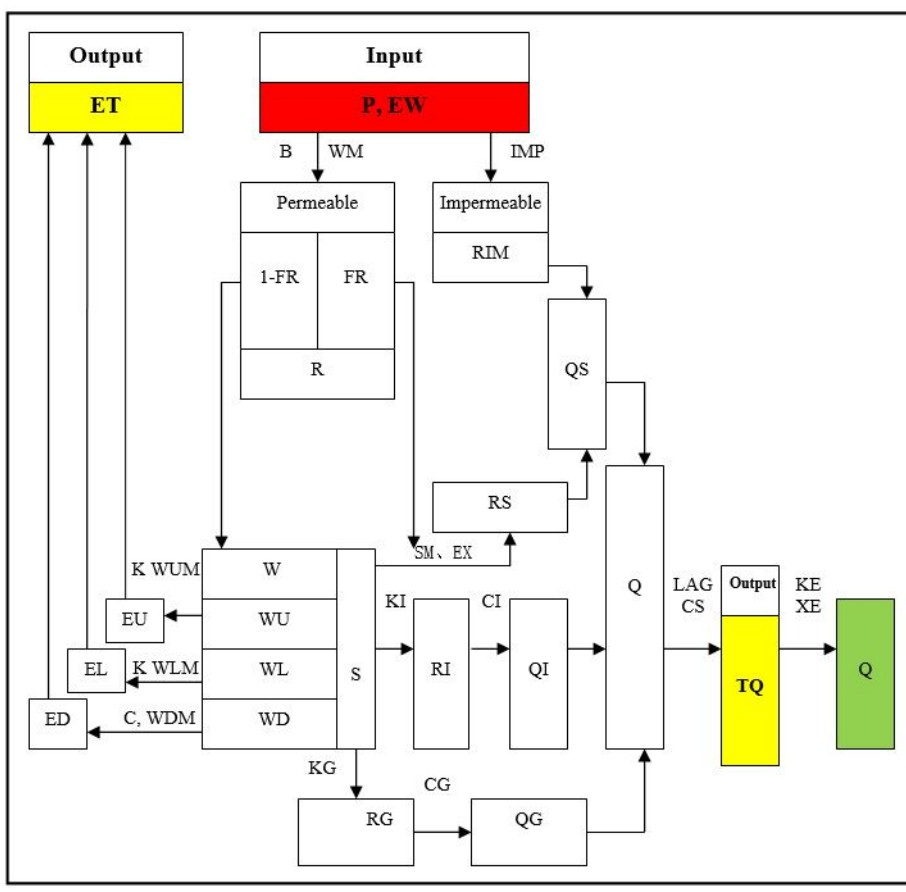

**Figure 1.** Flow chart of the XAJ model. The red box denotes the model input consisting of precipitation(P) and pan evaporation (EW); the yellow boxes denote the model simulated fluxes including evapotranspiration (ET) and discharge (TQ); the green box denotes the discharge after routing simulation (usually using the Muskingum method). Symbols outside the boxes are model parameters.





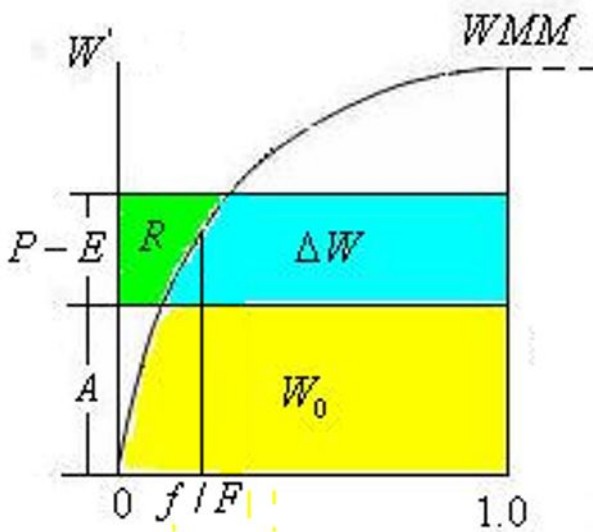

**Figure 2.** Tension water capacity curve of the XAJ model, which describes the tension water capacity distribution over the catchment. Each point on the curve represents a tension water capacity (Y axis, ranging from 0 to $WMM$) and the proportion of the area that has a tension water capacity no larger than that value (X axis). The mass balance calculation procedures of the XAJ model are presented: for a given time step when precipitation (P) is larger than evapotranspiration (ET), the difference between P and Evapotranspiration (ET) is partitioned into runoff (R, green shaded area) and soil water ($\Delta$W, blue shaded area) based on the tension water capacity curve and initial soil water ($W_0$, yellow shaded area)





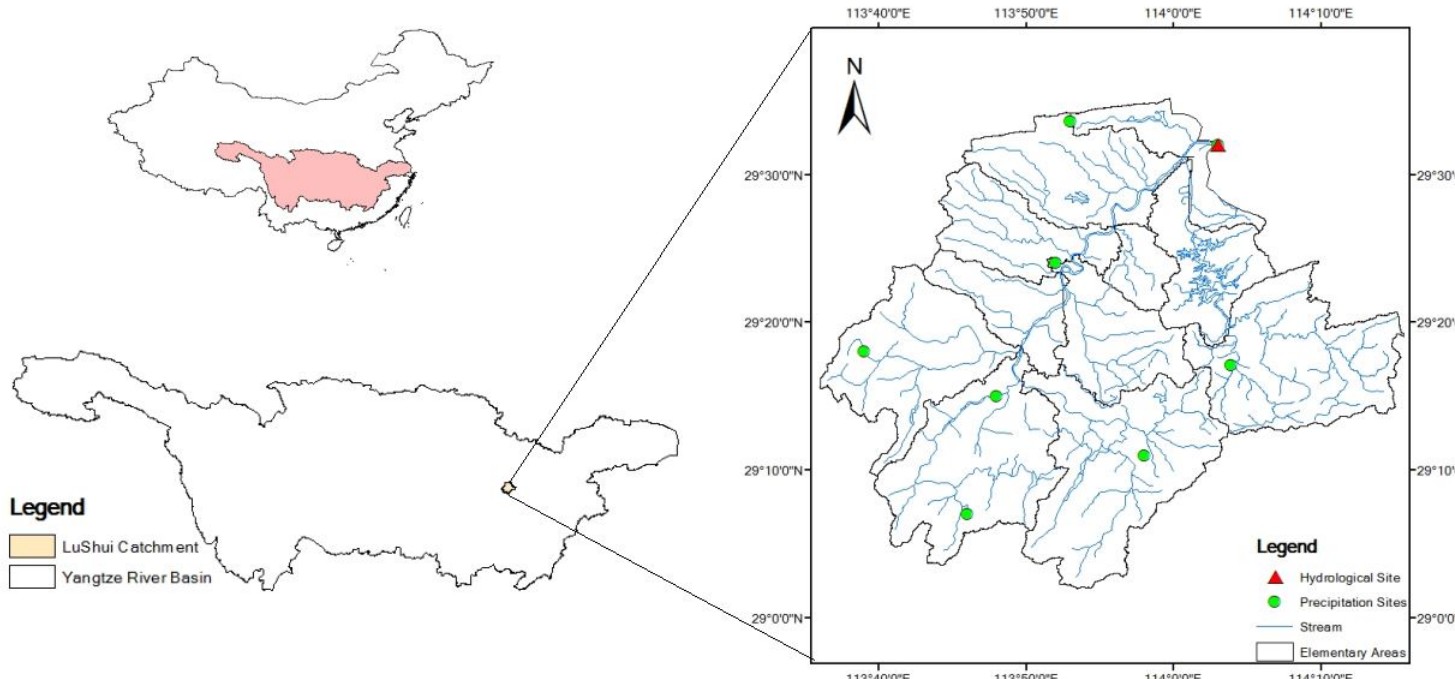

**Figure 3.** Lushui (LS) catchment. The red triangle is the ChongYang hydrological site that observes precipitation, pan evaporation and runoff; the green dots are precipitation sites set up to measure precipitation only; the blue line is the LS stream. The elementary areas are also shown, which are the computation unit of the XAJ models (see Section 2.1 for details).





**Figure 4.** Comparison between observed and XAJ-modeled daily runoff for the ChongYang hydrological site for the calibration (a; Jan 1, 2004–Dec 31, 2005) and validation (b; Jan 1, 2006–Dec 31, 2007) period. Also shown the evaluation metrics including: RMSE, NSE, and bias.





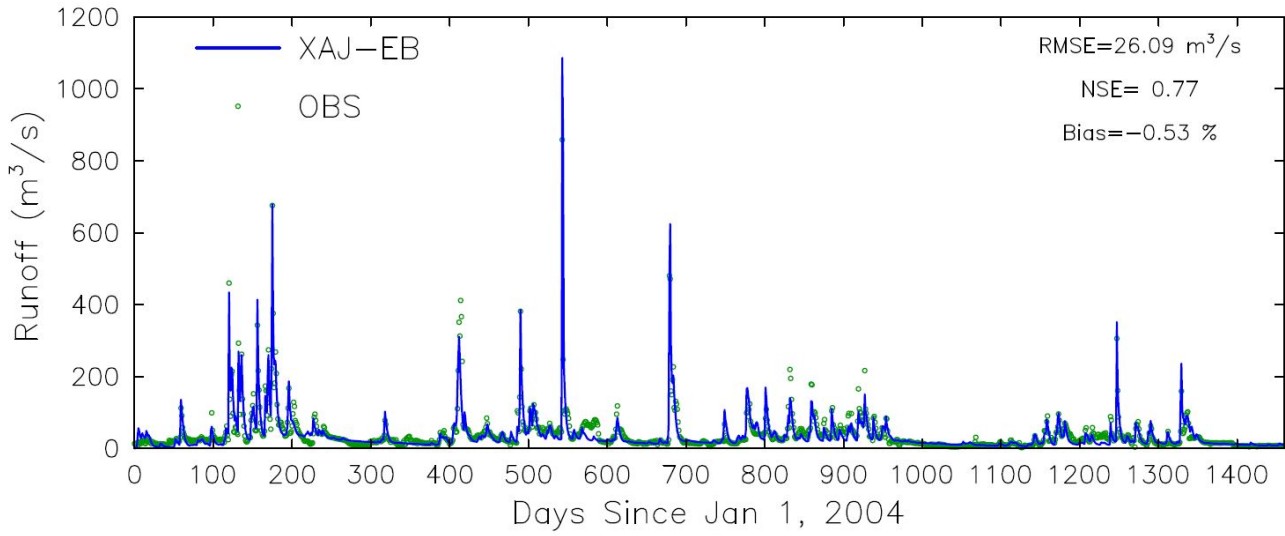

**Figure 5.** Comparison between observed and XAJ-EB modeled daily runoff for tue ChongYang hydrological site for period between Jan 1, 2004–Dec 31, 2007. Also shown the evaluation metrics including: RMSE, NSE, and bias.





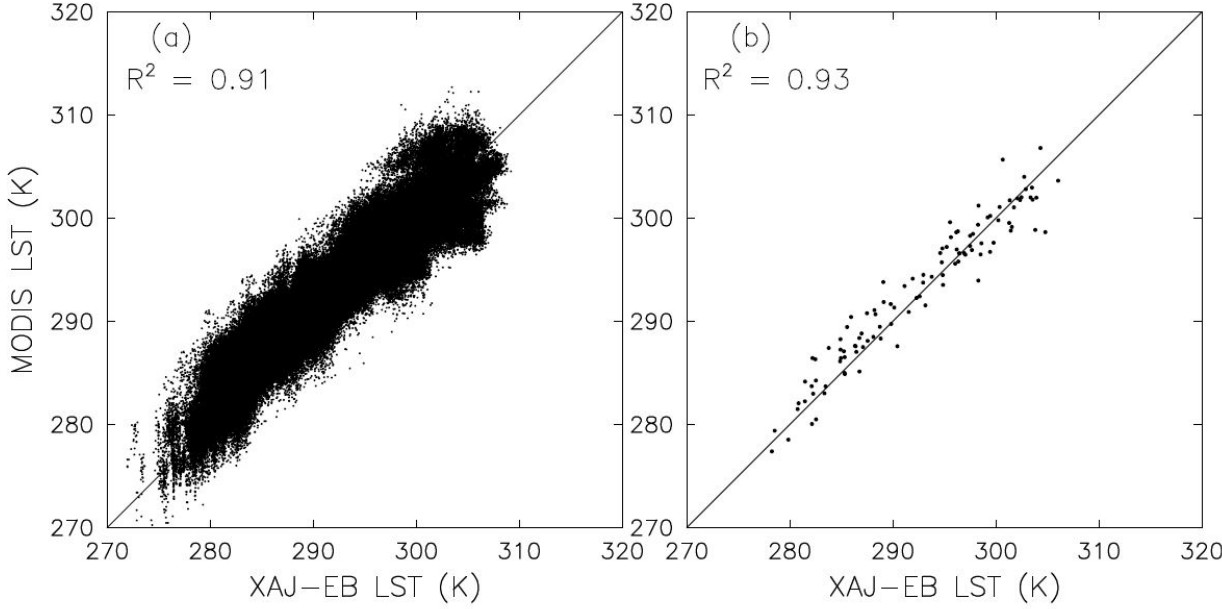

**Figure 6.** Scatter plot of MODIS retrieved and XAJ-EB modeled land surface temperature (LST), (a): catchment average LST and (b): grid LST.





**Figure 7.** Time series of modeled catchment average land surface temperature for the period between Jan 1, 2004–Dec 31, 2007. The MODIS-retrieved catchment-average land surface temperature (LST) is also shown when the data availability exceeded the threshold(see Section 3 for details).





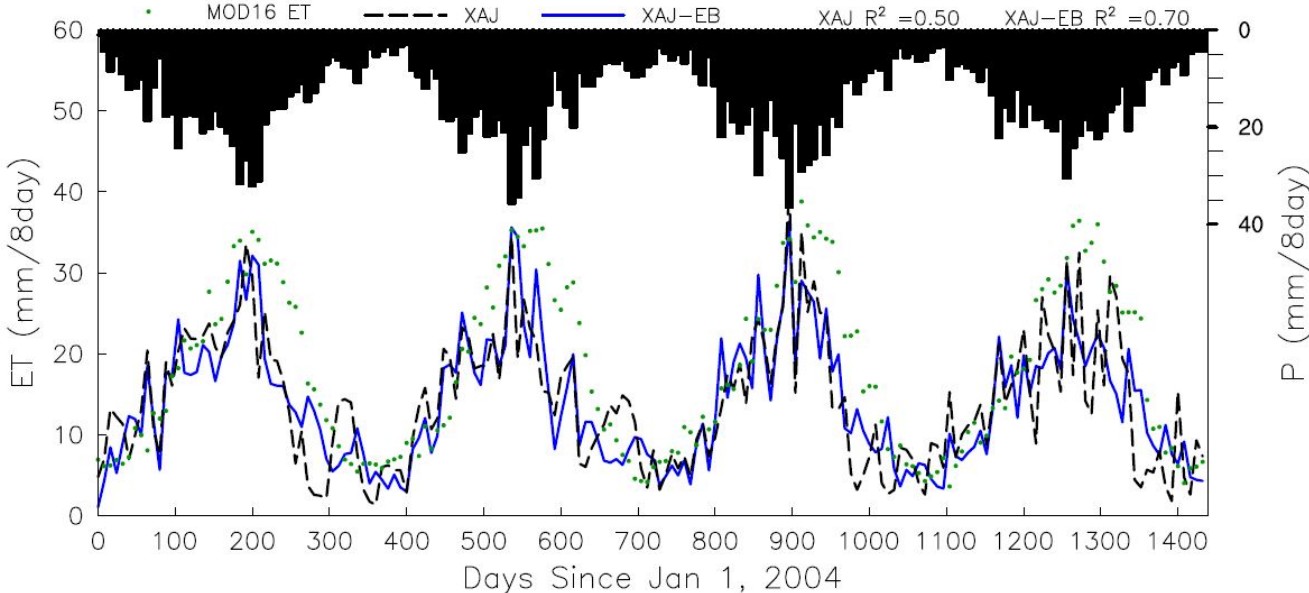

**Figure 8.** Comparison among MODIS-retrieved, XAJ and XAJ-EB-modeled 8-day catchment-average evapotranspiration (ET) for period between Jan 1, 2004–Dec 31, 2007. The 8-day gauge-measured catchment-average precipitation is also shown for the same period.





**Figure 9.** Comparison between XAJ and XAJ-EB modeled daily runoff (a) and soil moisture (b) for the yea 2007, soil moisture is expressed as equivalent water depth (soil depth * volumetric soil moisture) as defined by the XAJ model.





**Figure 10.** Comparison between XAJ-EB and Noah-MP modeled catchment average latent heat flux (a) and land surface temperature(b). (c) illustrates the comparison between catchment average precipitation used by XAJ-EB and Noah-MP.





**Table 1.** Parameter symbols and corresponding definitions in the XAJ model, Also shown the parameters calibrated for Lushui(LS) catchment during year 2004 and 2005.

| Parameter | Definition | Calibrated Value for the LS Catchment |
|-----------|------------|---------------------------------------|
| $IMP$ | Ratio of impermeable area to the total area in the catchment | 0.01 |
| $KC$ | Ratio of potential evapotranspiration to pan evaporation | 0.95 |
| $B$ | Exponential of the distribution of tension water capacity | 0.50 |
| $WM$ | Elementary area mean tension water capacity | Vary |
| $WUM$ | Tension water capacity of upper layer | Vary |
| $WLM$ | Tension water capacity of lower layer | Vary |
| $C$ | Deeper evapotranspiration coefficient | 0.10 |
| $SM$ | Elementary area mean free water capacity | 45.00 |
| $EX$ | Exponential of the distribution of free water capacity | 1.50 |
| $KG$ | Outflow coefficient of free water storage to the ground flow | 0.40 |
| $KI$ | Outflow coefficient of free water storage to the sub-flow | 0.40 |
| $CI$ | Recession constant of sub-flow storage | 0.63 |
| $CG$ | Recession constant of groundwater storage | 0.99 |
| $CS$ | Recession constant during stream routing in the elementary area | 0.55 |
| $LAG$ | Lag time step during stream routing in the elementary area | 0 |





**Table 2.** Variable, MODIS product name and spatial and temporal resolution of the remote sensing data used in this paper

| Remote Sensing Variable | MODIS Product Name | Spatial Resolution | Temporal Resolution |
| --- | --- | --- | --- |
| LAI | MCD15A3 | 1 km | 4-day |
| Albedo | MCD43A1 | 500 m | 16-day |
| LST | MOD11A1 | 1 km | Daily |
| ET | MOD16A2 | 1 km | 8-day |