# Peer review of "Improving the Xin'anjiang Hydrological Model Based on Mass-Energy Balance"

_Hydrology and Earth System Sciences, 2016_

## Referee Comment (RC1) · J. Ball (Referee) · 12 Jan 2017

The authors should be congratulated on production of an interesting paper that is easy to read and comprehend. There are a number of points, however, where the authors may wish to expand upon their current explanation. These include: 1 - Both the Abstract and the Introduction highlight the use of the model as a flood forecasting system. Hence only high flows are of interest with low flows being those between flood periods and of interest only for initializing the high flow periods. Nonetheless, calibration and validation of the models is focused on the capacity of the system to reproduce complete time-series of flows. Hence, this point is focused on whether reproduction of flow time-series is the appropriate metric for a flood forecasting modelling system.

The emphasis on reproduction of complete time-series of flows further complicates the

calibration as there will be more high flows than low flows potentially resulting in a bias towards reproduction of low flow situations. It would be worthwhile to consider how the high flows were simulated in the absence of any low flows; in other words, if the flow time-series were censored to remove the low flows, would the calibration and validation statistics be changed.

It is worth noting that recent publications (see, for example, Sakal et al., 2016) have shown that, for many modelling systems, different parameter sets are needed for adequate reproduction of both high and low flow time-series. Use of alternative parameter sets and the selection between them becomes a question then of the belief in the suitability of the model for forecasting of flows in a given regime; this belief can be expressed as a probability if desired.

2 - A second issue related to the calibration and validation of the modelling system relates to the input data and the parameters. As outlined in the paper, the XAJ model assumes homogeneity of the element area. Variability within the element area for those processes associated with conversion of rainfall to runoff is considered as discussed in the paper. However, the variability in the rainfall over the full catchment area is not discussed and consequently, variability of rainfall over the element area is not discussed.

If, as suspected from interpretation of the paper, rainfall at the centroid of the element area was used as input to element, then there is a need to describe how the rainfall was estimated. This estimation of rainfall depth will introduce an error into the modelling system. The likely magnitude of the error in rainfall estimation was discussed by Ball and Luk (1998) who, for a smaller catchment, showed the likely magnitude of errors in rainfall estimation using a number of alternative spatial interpolation schemes. Subsequent studies by, for example Mandapaka et al. (2009) and Younger et al. (2009) have considered the effects of rainfall uncertainty on catchment modelling outcomes.

The relevant point here is the need to ensure that the enhanced XAJ model outcomes are better than the XAJ model outcomes and that the uncertainty arising from the

rainfall inputs does not mask changes in the forecast errors.

References

Ball, JE and Luk, KC, (1998), Modelling the spatial variability of rainfall over a catchment, ASCE, Journal of Hydrologic Engineering, 3(2):122-130.

Mandapaka, PV, Krajewski, WF, Mantilla, R, and Gupta, VJ, (2009), Dissecting the effect of rainfall variability on the statistical structure of peak flows, Advances in Water Resources, 32(10):1508-1525

Sakal, A, Ball, JE and van Kalken, T, (2016), Concept of an Integrated Hydrological Ensemble Prediction System, Journal of Applied Water Engineering and Research, DOI 10.1080/23249676.2016.1224690

Younger, PM, Freer, JE, and Beven, KJ, (2009), Detecting the effects of spatial variability of rainfall on hydrological modelling within an uncertainty analysis framework, Hydrological Processes, 23(4), DOI: 10.1002/hyp.7341
* * *

---

## Referee Comment (RC2) · Anonymous Referee #2 · 6 Feb 2017

This paper is well written and has presented significant improvement to the Xin'Anjiang Model. The original XAJ model has been widely applied in flood forecasting in South part of China and has provided reasonable results. However, the XAJ model has known issues in parameterizing some of the important variables in mass balance. Moreover the original XAJ model failed to model the energy exchange of water, e.g. the ET was not modeled physically. This paper have made significantly improvements in the energy flux components parameterization and thus physically simulated the ET using the widely available remote sensing data nowadays. From the modeling results, the improvements are promising and are acceptable into operational applications in flood forecasting. The improvements also extend the applicability of the XAJ model which may be applied into arid or semi-arid area although more work is needed before proven. This reader suggest the authors to conduct more testing work beyond this paper in the

future, which will demonstrate how the XAJ-EB performs in different circumstances. Although the paper is well prepared, this reader suggest the author to carefully edit the paper before accepted for publications.

Specific issues:

1) Add spaces before and after some equations. The spacing needs to be consistent throughout the paper.

2) Carefully edit the paper. Space needed between some sentences which can be easily found.

3) I would like to suggest the authors to use a title like " Improving the XAJ hydrological model by adding energy simulation". The major improvements were made to simulating the energy flux and have not made significant change to the mass balance.

4) page 1 line 15: Change "application fields" into applicability.

5) page 2 line7: add ',″ before and

6) page 3 line 1: change should be into needs.

7) page 3 line 22: delete by

8) page 8 line 24: Have you considered the time difference of water concentration of different cells in the aggregation procedures?

9) page 13, line 3: change plotlted into plotted

10) page 14 line 15: change "," before Moreover into "."

11) page 15 line 5-8: simplify this sentence. This sentence is too complicate and very confusing for reader to understand.

12) page 15 line 21: change solved into simulated

13) page 15 line28: change " matched with the observed runoff well" into "well matched

the observed runoff"

14) page 16 line 4: change can help into helps

15) page 16 line 6: delete " may help to"

17) this reader would like to suggest authors to minimize the uses of "we" in this paper.

---

## Author Comment (AC1) · 9 Apr 2017

Dear Dr. Ball,

We greatly appreciate your thoughtful comments and suggestions that helped us to improve the manuscript. Based on your comments, we have revised the manuscript accordingly and we believe that your comments have been addressed. In the following, we give a point-by-point reply to your comments.

**(1) Reply to comment 1 regarding model calibration**

Thanks for this critical and good comment. We agreed that high flows are more important in real-time flood forecasting. Actually the reason why we performed the two-step calibration (Section 3.4) is to better reproduce the high flows. We first calibrated the parameters based on the complete time-series of runoff observations using PEST, which fits both low and high flows. Then for the second step, we adjusted the parameters by trial and error approach, mainly according to high flows. Such straightforward but effective calibration strategy stroked a balance between the representativeness of the runoff data and the importance of large flood events, providing the reliable baseline to validate the XAJ-EB model in this study. We have revised the Section 3.4 to describe the calibration strategy we used clearly.

Moreover, by coupling the mass balance, we added more constrains to the XAJ model, which can help the model to reproduce more reliable hydrological processes in both flood and non-flood periods. As such, the complete time-series of runoff observations are necessary in order to evaluate Runoff/ET/LST from XAJ-EB under various hydrometeorological conditions. We also revised the introduction part to better justify our motivations.

We also added a separate section regarding the parameters calibration to address your concern (Section 4.3). Generally, runoff data including both dry and wet conditions is required to represent the various characteristics of the catchment, from which the stable and robust parameter values can be obtained (PERRIN et al., 2007; Razavi and Tolson, 2013; Singh and Bárdossy, 2012) . However, short-period runoff observations of wet period can be used for calibration if the data availability is poor, which is also able to provide acceptable calibration results (Kim and Kaluarachchi, 2009; Sun et al., 2016; Wang et al., 2017). In this study, the complete time-series of runoff observations are necessary, however, a more rigorous quantification of the uncertainties in parameters calibration is need for the XAJ and XAJ-EB model in subsequent studies.

**(2) Reply to comment 2 regarding precipitation interpolation**

Thanks for this question. The spatially-distributed precipitation is included in the meteorological forcing data, however, after evaluating against precipitation measured by gauges, we found that spatially-distributed precipitation is biased (Fang et al., 2017) . For this reason, we used the precipitation from 8 gauges instead.

Several approaches existed for spatially interpolating the precipitation from gauges, however, their performances and uncertainties depend on certain conditions including the pattern of precipitation, the characteristic of catchment, and the locations of gauges (Ball and Luk, 1998; Di Piazza et al., 2011; Zhang and Srinivasan, 2009).   It's difficult to evaluate the performance of different interpolation approaches in LS since the true areal precipitation is theoretically not available. In this paper, the conventional Thiessen polygon approach was employed, this is because:

1) Thiessen polygon has been intensively used in the XAJ model for flood simulation and foresting (Xia and Zhang, 2009)

2) The potential uncertainties raising from the precipitation interpolation are not the major concern of this paper. Actually each interpolation approach can lead to uncertainties, which is difficult to evaluate in the study area since the true areal precipitation is theoretically not available.

We added the description of how we interpolate precipitation in the revised manuscript (Page 10 Line 30). The precipitation of gauges are interpolated to grids by following steps:

1) Thiessen polygons are generated according to the geographic locations of gauges;

2) Thiessen polygons are overlaid with element areas, and the precipitation of ith element area (Pi) is weighted by Thiessen polygons that intersected with it (the precipitation of green-filled element area in Figure 1 is determined by Thiessen polygons 1, 3, 4 and 5, taking their area as weight);

3) Grids belongs to the same element area i are assigned the same precipitation Pi.

using such scheme we can ensure that the areal mean precipitation of element area of both   XAJ-EB and XAJ model are the same, and therefore the differences between two model are not result from the precipitation differences.

Xingnan Zhang
On behalf of all co-authors

[Figure]

**Figure 1. Sketch plot of precipitation calculation based on element areas and Thiessen polygons.**

**Reference**

Ball, J. E. and Luk, K. C.: Modeling Spatial Variability of Rainfall over a Catchment, J. Hydrol. Eng., 3(2), 122–130, doi:10.1061/(ASCE)1084-0699(1998)3:2(122), 1998.

Fang, Y.-H., Zhang, X., Niu, G.-Y., Zeng, W., Zhu, J. and Zhang, T.: Study of the Spatiotemporal Characteristics of Meltwater Contribution to the Total Runoff in the Upper Changjiang River Basin, Water, 9(3), 165, doi:10.3390/w9030165, 2017.

Kim, U. and Kaluarachchi, J. J.: Hydrologic model calibration using discontinuous data: an example from the upper Blue Nile River Basin of Ethiopia, Hydrol. Process., n/a–n/a, doi:10.1002/hyp.7465, 2009.

PERRIN, C., OUDIN, L., ANDREASSIAN, V., ROJAS-SERNA, C., MICHEL, C. and MATHEVET, T.: Impact of limited streamflow data on the efficiency and the parameters of rainfall—runoff models, Hydrol. Sci. J., 52(1), 131–151, doi:10.1623/hysj.52.1.131, 2007.

Di Piazza, A., Conti, F. Lo, Noto, L. V., Viola, F. and La Loggia, G.: Comparative analysis of different techniques for spatial interpolation of rainfall data to create a serially complete monthly time series of precipitation for Sicily, Italy, Int. J. Appl. Earth Obs. Geoinf., 13(3), 396–408, doi:10.1016/j.jag.2011.01.005, 2011.

Razavi, S. and Tolson, B. A.: An efficient framework for hydrologic model calibration on long data periods, Water Resour. Res., 49(12), 8418–8431, doi:10.1002/2012WR013442, 2013.

Singh, S. K. and Bárdossy, A.: Calibration of hydrological models on hydrologically unusual events, Adv. Water Resour., 38, 81–91, doi:10.1016/j.advwatres.2011.12.006, 2012.

Sun, W., Wang, Y., Cui, X., Yu, J., Zuo, D. and Xu, Z.: Physically-based distributed hydrological model calibration based on a short period of streamflow data: case studies in two Chinese basins, Hydrol. Earth Syst. Sci. Discuss., (May), 1–20, doi:10.5194/hess-2016-192, 2016.

Wang, L., van Meerveld, H. J. and Seibert, J.: When should stream water be sampled to be most informative for event-based, multi-criteria model calibration?, Hydrol. Res., nh2017197, doi:10.2166/nh.2017.197, 2017.

Xia, D. and Zhang, X.: Construction pattern of distributed real-time flood forecast schemes (In Chinese), J. Hohai Univ. (Natural Sci., 37(5), 516–522, 2009.

Zhang, X. and Srinivasan, R.: GIS-Based Spatial Precipitation Estimation: A Comparison of Geostatistical Approaches, JAWRA J. Am. Water Resour. Assoc., 45(4), 894–906, doi:10.1111/j.1752-1688.2009.00335.x, 2009.

---

## Author Comment (AC2) · 9 Apr 2017

Dear Reviewer:

We greatly appreciate your thoughtful comments and suggestions that helped us to improve the manuscript. Based on your comments, we have revised the manuscript accordingly and we believe that your comments have been addressed. In the following, we give a point-by-point reply to your comments.

Specific Comments:

**(1) Add spaces before and after some equations. The spacing needs to be consistent throughout the paper.**

**Response:** We used LaTex instead of WORD to prepare this manuscript, and under such circumstance the layouts of these equations are automatically controlled by the official LaTex template. It seems that, by default, the template places equations on the leftmost edge and places corresponding numbers on the rightmost edge, which results in inconsistent spacing. We can not adjust the layouts since we're not allowed to modify the original class file. However, we found that we aligned several equations (Eq.14; Eq.25 - 29) based on equal sign "=" which generated some spacing before these equations. In the revised manuscript, we have canceled the alignment to remove the spacing.

**(2) Carefully edit the paper. Space needed between some sentences which can be easily found.**

**Response:** Like equations, both line spacing and paragraph spacing are controlled automatically by LaTex template. After checking on the manuscript, we found that the line spacing of Page 4 Line17-25 is larger. This is because we used the enumerate environment to organize this part, and in the revised manuscript we used the ordinary method instead, which helps to reduce the line spacing.

**(3) I would like to suggest the authors to use a title like " Improving the XAJ hydrological model by adding energy simulation". The major improvements were made to simulating the energy flux and have not made significant change to the mass balance.**

**Response:** We are sorry for the misunderstanding we raised. The novelty of this study include:

1) Development of the energy balance scheme suitable for the XAJ model

2) Fully coupling of the mass and energy balance in the model

We have revised the Section 2.2.1 to explain the methodology more clearly. Moreover, we also revised the introduction part to better justify our motivations.

**(4) page 1 line 15: Change "application fields" into applicability.**

**Response:** We have changed "*application fields*" into "*applicability*" in the revised manuscript.

**(5) page 2 line7: add '," before and**

**Response:** We have added comma before "*and*" in the revised manuscript.

**(6) page 3 line 1: change should be into needs.**

**Response:** We used "needs" to replace "should be" in the revised manuscript.

**(7) page 3 line 22: delete by**

**Response:** We have deleted "by" in the revised manuscript.

**(8) page 8 line 24: Have you considered the time difference of water concentration of different cells in the aggregation procedures?**

**Response:** We didn't take consideration the time difference of water routing when aggregating the runoff yield. The time difference you mentioned is considered, at each element area, in routing simulation by several liner-reservoir components, which represents the "lumped" effects.

**(9) page 13, line 3: change plotlted into plotted**

**Response:** Sorry for the typo and we have changed the word "plotlted" into "plotted".

**(10) page 14 line 15: change "," before Moreover into "."**

**Response:** We have changed the "," into ".".

**(11) page 15 line 5-8: simplify this sentence. This sentence is too complicate and very confusing for reader to understand.**

**Response:** We have simplified the sentence.

**(12) page 15 line 21: change solved into simulated**

**Response:** We have changed "*solved* " into "*simulated*" in the revised manuscript.

**(13) page 15 line28: change " matched with the observed runoff well" into "well matched the observed runoff"**

**Response:** We have changed " *matched with the observed runoff well* " into "*well matched the observed runoff*" in the revised manuscript.

**(14) page 16 line 4: change can help into helps**

**Response:** We have changed " *can help* " into "*helps*" in the revised manuscript.

**(15)** page 16 line 6: delete " may help to"

**Response:** We have deleted " *may help to*".

**(16) this reader would like to suggest authors to minimize the uses of "we" in this paper.**

**Response:** We have deleted several "*we*" in the revised manuscript.

Xingnan Zhang
On behalf of all co-authors

---

## Author Comment (AC3) · 9 Apr 2017

**Improving the Xin'anjiang Hydrological Model Based on Mass-Energy Balance**

Yuan-Hao Fang[1,2,4], Xingnan Zhang[1,2,3], Chiara Corbari[4], Marco Mancini[4], Guo-Yue Niu[5,6], and Wenzhi Zeng[7,8]

[1]National Cooperative Innovation Center for Water Safety & Hydro-Science, Hohai University, Nanjing, China
[2]College of Hydrology and Water Resources, Hohai University, Nanjing, China
[3]National Engineering Research Center of Water Resources Efficient Utilization and Engineering Safety, Hohai University, Nanjing, China
[4]Department of Civil and Environmental Engineering (D.I.C.A.), Politecnico di Milano, Milan, Italy
[5]Department of Hydrology and Atmospheric Sciences, University of Arizona, Tucson AZ, U.S.A
[6]Biosphere 2, University of Arizona, Oracle AZ, USA
[7]State Key Laboratory of Water Resources and Hydropower Engineering Science, Wuhan University, Wuhan, China
[8]Crop Science Group, Institute of Crop Science and Resource Conservation (INRES), University of Bonn, Bonn, Germany

[revised manuscript text omitted]

At present, however, the traditional calibration approach is becoming more challenging for the XAJ model. This is partly due to the fact that the model parameters have become distributed to take account of the heterogeneities of the catchment (Xia and

20   Zhang, 2009; Yao et al., 2012), which theoretically requires more constraints to calibrate and validate these spatially-distributed parameters.

Moreover, although precipitation and runoff can be measured by traditional approaches, accurate catchment average ET and SMC are difficult to obtain at the catchment scale. Therefore the bias of simulations is more likely to be accumulated in SMC when only mass balance is considered. The significance of SMC to real-time flood forecasting has gradually been recognized.

[revised manuscript text omitted]

The estimation of areal mean precipitation in the computational grid is crucial for the accurate hydrological modeling. Several approaches existed for spatially interpolating the precipitation from gauges, however, their performances and uncertainties

depend on certain conditions including the pattern of precipitation, the characteristic of catchment, and the locations of gauges (Ball and Luk, 1998; Di Piazza et al., 2011; Zhang and Srinivasan, 2009). It's difficult to evaluate the performance of different interpolation approaches in LS since the true areal precipitation is theoretically not available. In this paper, the conventional Thiessen polygon approach, the one intensively used in the XAJ model, was employed to derive the spatially-distributed pre-

5    cipitation from 8 gauges. To make the precipitation inputs of XAJ-EB comparable to those of XAJ, the precipitation of gauges are interpolated to grids by following steps:

(1) Thiessen polygons were generated according to the geographic locations of gauges;

(2) Thiessen polygons were overlaid with element areas, and the precipitation of $i$th element area ($P_i$) was weighted by Thiessen polygons that intersected with it (the precipitation of green-filled element area in Figure 4 was determined by Thiessen

10    polygons 1, 3, 4 and 5, taking their area as weight);

(3) Grids belongs to the same element area $i$ were assigned the same precipitation $P_i$.

As seen from Eq. 4 through Eq. 22, a range of parameters/variables is needed for the energy balance scheme. Each grid was assigned a set of time-independent parameters including soil physical properties (e.g. $\theta_{fc}$) , vegetation properties (e.g. $r_{s\ min}$) based on soil and vegetation type. Other time-dependent variables were obtained either from remote sensing data (e.g. LAI) or

15    model simulated states (e.g. $\theta$).

**3.4    Calibration and Validation of the XAJ Model**

As listed in Table 1, several parameters have to be estimated before applying the XAJ model, among which tension water capacity $WM$ has physical definition that can be estimated from the soil proprieties for each grid:

$$WM = (\theta_{fc} - \theta_{wp}) \times SD \tag{24}$$

20    where SD is soil depth (mm). All three soil proprieties can be retrieved from soil dataset described above.

For parameters other than $WM$, two years of data (2004 and 2005) was chose to perform a calibration against observed runoff data using the original XAJ model, which calculates ET using measured pan evaporation.

We first introduced a model-independent parameter estimation tool, namely PEST (Doherty et al., 1994) to provide an optimized combination of parameters. PEST is based on the Gauss-Marquardt Levenberg (GML) algorithm (Marquardt, 1963)

25    and has been widely applied in calibrating hydrological models. The initial values as well as the optimization limits of the parameters were set according to Zhao (1984). After the automatic calibration, we used the traditional trial and error method to adjust some parameters based on our experience in calibrating the XAJ model. This is because the algorithm implemented by PEST tries to fit the complete time-series of runoff observations, regardless of high flows or low flows, while the flood events are the major concerns of real-time flood forecasting. As such, the trial and error method was applied to improve the

30    simulations of high flows based on parameters optimized by PEST.

[revised manuscript text omitted]
 the simulated upward long wave radiation which represents the integration of different land surface components within the grid (see Niu et al., 2011, for details). Figure 11a shows a generally good agreement between the XAJ-EB and Noah-MP simulated LST, with RMSE, NSE and bias values as 1.68 K, 0.97 and -0.20%. Such good agreement indicates that, comparing with Noah-MP with multiple LSTs, the energy balance scheme of XAJ-EB is able to produce reliable LST with only one lumped temperature.

As for the latent heat flux, although the overall bias was small (-2.53 %), low NSE (0.53) indicates there is an inconsistency in inconsistence of the ET time series. This is partly due to the different precipitation fields we used for the 2 models, which had an NSE of only 0.51 (Figure 11c). By comparing Figure 11b with Figure 11c we found that a larger bias of ET generally corresponds to a larger bias in precipitation.

**Figure 11 here**

In addition to the simplification of the energy balance scheme, XAJ-EB also makes use of various remote sensing products (i.e. LAI, Albedo, etc.) to eliminate the processes that have little effects on flood forecasting (e.g. vegetation dynamics), only retaining the essential processes that related to ET and runoff simulation, which help to reduce both the complexity of the model and number of parameters need calibrated.

**4.3 Calibration strategy of the model**

Calibration is necessary for hydrological models, even for the physically-based models (Singh and Bárdossy, 2012). There has been the concern regarding the selection of runoff observations for calibration. In general, runoff data including both dry and

wet conditions is required to represent the various characteristics of the catchment, from which the stable and robust parameter values can be obtained (PERRIN et al., 2007; Razavi and Tolson, 2013; Singh and Bárdossy, 2012). However, considering the data availability, researchers have been studying the calibration strategies using runoff observations of short period (e.g. Kim and Kaluarachchi, 2009; Sun et al., 2016; Wang et al., 2017). Although specific results differ depending on the catchments as well as the models adopted, they all found that high-flow periods exert greater influence on model calibration, which implies that the hydrological models can be calibrated against high flows, which are more important for real-time forecasting.

To validate the energy balance scheme as well as the XAJ-EB model developed in this paper, a two-step calibration of the XAJ model was applied (Section 3.4). The first step was to calibrate the parameters using PEST based on the complete time-series of runoff observations. This is because the calibration of parameter $KC$ requires complete runoff observations of several years to ensure that the accumulated simulated runoff was close to the corresponding observed value (Zhao, 1992); and (2) the validation of LST and ET from XAJ-EB also requires accurate mass balance simulations all over the year, covering flood and non-flood periods, to better evaluate the performance of the model under various hydrometeorological conditions. Then for the second step, the trial and error method was applied to adjust the resulting optimized parameters, mainly according to the high flows of flood events. Such straightforward but effective calibration strategy stroked a balance between the representativeness of the runoff data and the importance of large flood events, providing the reliable baseline to validate the XAJ-EB model in this study. However, a more rigorous quantification of the uncertainties in parameters calibration is need for the XAJ and XAJ-EB model in subsequent studies.

**5   Conclusion**

[revised manuscript text omitted]

Wang, L., van Meerveld, H. J., and Seibert, J.: When should stream water be sampled to be most informative for event-based, multi-criteria model calibration?, Hydrol. Res., p. nh2017197, doi:10.2166/nh.2017.197, http://hr.iwaponline.com/content/early/2017/03/13/nh.2017.197.abstracthttp://hr.iwaponline.com/lookup/doi/10.2166/nh.2017.197, 2017.

Xia, D. and Zhang, X.: Construction pattern of distributed real-time flood forecast schemes (In Chinese), J. Hohai Univ. (Natural Sci., 37, 516–522, 2009.

[revised manuscript text omitted]

---

## Author Comment (AC4) · 9 Apr 2017

**Improving the Xin'anjiang Hydrological Model Based on Mass-Energy Balance**

Yuan-Hao Fang[1,2,4], Xingnan Zhang[1,2,3], Chiara Corbari[4], Marco Mancini[4], Guo-Yue Niu[5,6], and Wenzhi Zeng[7,8]

[1]National Cooperative Innovation Center for Water Safety & Hydro-Science, Hohai University, Nanjing, China
[2]College of Hydrology and Water Resources, Hohai University, Nanjing, China
[3]National Engineering Research Center of Water Resources Efficient Utilization and Engineering Safety, Hohai University, Nanjing, China
[4]Department of Civil and Environmental Engineering (D.I.C.A.), Politecnico di Milano, Milan, Italy
[5]Department of Hydrology and Atmospheric Sciences, University of Arizona, Tucson AZ, U.S.A
[6]Biosphere 2, University of Arizona, Oracle AZ, USA
[7]State Key Laboratory of Water Resources and Hydropower Engineering Science, Wuhan University, Wuhan, China
[8]Crop Science Group, Institute of Crop Science and Resource Conservation (INRES), University of Bonn, Bonn, Germany

[revised manuscript text omitted]

 At present, however, the traditional calibration approach is becoming more challenging for the XAJ model. This is partly due to the fact that the model parameters have become distributed to take account of the heterogeneities of the catchment (Xia and Zhang, 2009; Yao et al., 2012), which theoretically requires more constraints to calibrate and validate these spatially-distributed parameters.

Moreover, although precipitation and runoff can be measured by traditional approaches, accurate catchment average ET and SMC are difficult to obtain at the catchment scale. Therefore the bias of simulations is more likely to be accumulated in SMC  when only mass balance is considered. The significance of SMC to real-time flood forecasting has gradually been recognized. Studies showed that the bias in predicting flood peaks is related to unrealistic antecedent SMC estimation (Huza et al., 2014), and therefore the performance of real-time flood forecasting can be improved by setting or assimilating initial SMC (Brocca et al., 2009; Berthet et al., 2009; Komma et al., 2008; Tramblay et al., 2010; Wanders et al., 2014). The accuracy of SMC estimation before flood events largely depends on ET estimation. In addition, considering the abilities to extend the leading time and quantify predictability, ensemble flood forecasting techniques are more attractive today (Cloke and Pappenberger, 2009), and the estimation of SMC and ET is even more important in ensemble flood forecasting due to a longer leading time.

For the aforementioned reasons, it is therefore necessary to introduce more constraints to the XAJ model, and the energy balance can serve this purpose well since the hydrological processes are governed by both mass and energy balance. One feasible way to introduce the energy balance to the XAJ model is through ET. As is discussed before, the simple and empirical ET routine of the XAJ model is

5  based on mass balance only, and the major defects of the ET routine  are: (1) the input pan evaporation is measured only at few specific locations, reflecting daily evaporation from open water, which means that the potential ET (PET) over a large area is assumed to be the same. Such an assumption dose not always hold under heterogeneous meteorology or underlying surface conditions (Xu et al., 2006; Yuan et al., 2008); (2) calibration of $K_c$ (see Section 2.1 for details), a sensitive parameter

10 of the XAJ model controlling water balance, is needed to convert pan evaporation to PET, which is impossible for ungauged catchments where observed runoff is unavailable; and (3) the empirical relationship linking PET with actual ET only takes water balance into account, neglecting other factors (e.g. meteorological conditions) that control ET processes (Wang and Dickinson, 2012).

15 It  needs noted that the  energy balance-based ET schemes have been intensively studied in the land surface modeling community (Overgaard et al., 2006). Land surface models (LSMs) are developed to provide various fluxes and states connecting the atmosphere and land surface (Overgaard et al., 2006; Niu et al., 2011). Most LSMs have an energy balance component for ET estimation, but the way these models solve the energy balance differs. According to Su (2002) and Kalma et al. (2008), generally three different approaches are employed by LSMs for ET estimation: (1) calculate

20 all energy balance components except latent heat flux, which is obtained as the residual of the energy budget; (2) compute all components involved in energy balance by closing the balance equation, latent heat is solved at the same time when energy budget is closed; and (3) an empirical approach using water stress to derive ET. However, these approaches are rarely applied in hydrological models,especially for real-time flood forecasting, because their structures are complex and generally require considerable data and parameters to drive the model.

25 Benefiting from remote sensing and data assimilation techniques,  more meteorological and land surface data are available now. The scientific community has been working to improve ET scheme of hydrological models (e.g. Corbari et al., 2011; Niu et al., 2014; Spies et al., 2015; Yan et al., 2012). In particular, some efforts have been made to improve the ET simulation of the XAJ model. Methodologies reported can be summarized as two approaches. The first approach was to introduce a physically-based formula to simulate PET based on meteorological measurements, aiming to provide more accurate

30 PET input while the XAJ model structure remained unchanged (Yuan et al., 2008). The second approach involved replacing the ET routine by a more sophisticated scheme, typically the Penman-Monteith (PM) equation, which simulates actual ET by meteorological variables, remote sensing data and modeled SMC (Li et al., 2009; Zhou et al., 2013). These studies have demonstrated the feasibility of improving the ET scheme of the XAJ model. In these previous work, however, the mass balance and energy balance are either isolated or one-way linked, neglecting the interactions between them. Additionally,

35

 the PM equation employed tends to neglect evaporation due to the "big leaf" assumption (Yan et al., 2012).  A more scientific way to simulate ET is by  coupling both the mass and energy . For example, as reported by Corbari et al. (2011), a water balance model FEST (Rabuffetti et al., 2008) was augmented by coupling a energy balance scheme and various case studies have confirmed its applicability under different conditions (Masseroni et al., 2011; Corbari et al., 2013; Corbari and Mancini, 2014).

 The overall goal of this paper is to develop an energy balance scheme suitable for the XAJ model , with which the mass balance-based runoff yield scheme of the original XAJ model can be fully coupled. 
[revised manuscript text omitted]

The estimation of areal mean precipitation in the computational grid is crucial for the accurate hydrological modeling. Several approaches existed for spatially interpolating the precipitation from gauges, however, their performances and uncertainties depend on certain conditions including the pattern of precipitation, the characteristic of catchment, and the locations of gauges (Ball and Luk, 1998; Di Piazza et al., 2011; Zhang and Srinivasan, 2009). It's difficult to evaluate the performance of different interpolation approaches in LS since the true areal precipitation is theoretically not available. In this paper, the conventional Thiessen polygon approach, the one intensively used in the XAJ model, was employed to derive the spatially-distributed precipitation from 8 gauges. To make the precipitation inputs of XAJ-EB comparable to those of XAJ, the precipitation of gauges are interpolated to grids by following steps:

(1) Thiessen polygons were generated according to the geographic locations of gauges;

(2) Thiessen polygons were overlaid with element areas, and the precipitation of $i$th element area ($P_i$) was weighted by Thiessen polygons that intersected with it (the precipitation of green-filled element area in Figure 4 was determined by Thiessen polygons 1, 3, 4 and 5, taking their area as weight);

(3) Grids belongs to the same element area $i$ were assigned the same precipitation $P_i$.

As seen from Eq. 4 through Eq. 22, a range of parameters/variables is needed for the energy balance scheme. Each grid was assigned a set of time-independent parameters including soil physical properties (e.g. $\theta_{fc}$) , vegetation properties (e.g. $r_{s\ min}$) based on soil and vegetation type. Other time-dependent variables were obtained either from remote sensing data (e.g. LAI) or model simulated states (e.g. $\theta$).

**3.4 Calibration and Validation of the XAJ Model**

As listed in Table 1, several parameters have to be estimated before applying the XAJ model, among which tension water capacity $WM$ has physical definition that can be estimated from the soil proprieties for each grid:

$$WM = (\theta_{fc} - \theta_{wp}) \times SD \tag{24}$$

where SD is soil depth (mm). All three soil proprieties can be retrieved from soil dataset  described above.

For parameters other than $WM$,  two years of data (2004 and 2005) was chose to perform a calibration against observed runoff data using the original XAJ model, which calculates ET using measured pan evaporation.

We first introduced a model-independent parameter estimation tool, namely PEST (Doherty et al., 1994) to provide an optimized combination of parameters. PEST is based on the Gauss-Marquardt Levenberg (GML) algorithm (Marquardt, 1963) and has been widely applied in calibrating hydrological models. The initial values as well as the optimization limits of the parameters were set according to Zhao (1984). After the automatic calibration, we used the traditional trial and error method to adjust some parameters based on our experience in calibrating the XAJ model.  This is because the algorithm implemented by PEST tries to fit the complete time-series of runoff observations, regardless of high flows or low flows, while the flood events are the major concerns of real-time flood forecasting. As such, the trial and error method was applied to improve the simulations of high flows based on parameters optimized by PEST.

[revised manuscript text omitted]

30    represents the integration of different land surface components within the grid (see Niu et al., 2011, for d Figure 11a shows a generally good agreement between the XAJ-EB and Noah-MP simulated LST, with RMSE, NSE and bias values as 1.68 K, 0.97 and -0.20%compared with a more sophisticated LSM. Such good agreement indicates that, comparing with Noah-MP with multiple LSTs, the energy balance scheme of XAJ-EB is able to

produce reliable LST with only one lumped temperature, .

As for the latent heat flux, although the overall bias was small (-2.53 %), low NSE (0.53) indicates there is an inconsistency in inconsistence of the ET time series. This is partly due to the different precipitation fields we used for the 2 models, which had an NSE of only 0.51 (Figure 11c). By comparing Figure 11b with Figure 11c we found that a larger bias of ET generally corresponds to a larger bias in precipitation.

**Figure 11 here**

In addition to the simplification of the energy balance scheme, XAJ-EB also makes use of various remote sensing products (i.e. LAI, Albedo, etc.) to eliminate the processes that have little effects on flood forecasting (e.g. vegetation dynamics), only retaining the essential processes that related to ET and runoff simulation, which help to reduce both the complexity of the model and number of parameters need calibrated.

**4.3 Calibration strategy of the model**

Calibration is necessary for hydrological models, even for the physically-based models (Singh and Bárdossy, 2012). There has been the concern regarding the selection of runoff observations for calibration. In general, runoff data including both dry and wet conditions is required to represent the various characteristics of the catchment, from which the stable and robust parameter values can be obtained (PERRIN et al., 2007; Razavi and Tolson, 2013; Singh and Bárdossy, 2012). However, considering the data availability, researchers have been studying the calibration strategies using runoff observations of short period (e.g. Kim and Kaluarach Although specific results differ depending on the catchments as well as the models adopted, they all found that high-flow periods exert greater influence on model calibration, which implies that the hydrological models can be calibrated against high flows, which are more important for real-time forecasting.

To validate the energy balance scheme as well as the XAJ-EB model developed in this paper, a two-step calibration of the XAJ model was applied (Section 3.4). The first step was to calibrate the parameters using PEST based on the complete time-series of runoff observations. This is because the calibration of parameter $KC$ requires complete runoff observations of several years to ensure that the accumulated simulated runoff was close to the corresponding observed value (Zhao, 1992b); and (2) the validation of LST and ET from XAJ-EB also requires accurate mass balance simulations all over the year, covering flood and non-flood periods, to better evaluate the performance of the model under various hydrometeorological conditions. Then for the second step, the trial and error method was applied to adjust the resulting optimized parameters, mainly according to the high flows of flood events. Such straightforward but effective calibration strategy stroked a balance between the representativeness of the runoff data and the importance of large flood events, providing the reliable baseline to validate the XAJ-EB model in this study. However, a more rigorous quantification of the uncertainties in parameters calibration is need for the XAJ and XAJ-EB model in subsequent studies.

**5   Conclusion**

In this paper, an energy balance based scheme suitable for the XAJ model was developed by explicitly taking account of bare soil and the canopy using a "patch approach". Different energy fluxes for bare soil and the canopy respectively were parameterized. The energy balance was  simulated by determining RET, which is theoretically the LST that closes the balance. The energy balance scheme was then fully coupled to the mass mass balance scheme of the XAJ model. The improved model, XAJ-EB estimates the actual ET and runoff yield through a mass-energy balance approach using various meteorological and remotely sensed data.

Taking the LS catchment as the study area, we calibrated and validated the original XAJ model and used the same optimized parameters to evaluate the performance of the XAJ-EB model. With respect to the runoff simulation for the period between 2004 and 2007, the XAJ-EB model had RMSE, NSE and bias values of 26.09 m$^3$ s$^{-1}$, 0.77 and -0.53%, respectively. The results well matched with the observed runoff, which was also comparable to the original XAJ model. In addition to runoff, XAJ-EB is also capable of simulating the dynamics of LST with R$^2$, RMSE and NSE values of 0.93, 2.25 K and 0.89, respectively compared with MODIS-retrieved catchment average LST. The good match between modeled and remotely-sensed LST implies that the XAJ-EB model is able to reproduce the mass-energy balance processes since LST reflects the interactions among various processes. Moreover, with LST as an output, XAJ-EB adds a new constraint and offers a potentially new approach for model calibration and validation, especially when runoff data are unavailable.

The mass-energy balance scheme developed in this paper is comparable to the sophisticated LSM Noah-MP model in terms of LST and latent heat flux modeling, which overcomes several defects of the original XAJ model that simulates actual ET using pan evaporation measurements. The inter-comparison between XAJ-EB and XAJ shows that the improvement of ET estimation  helps to improve the runoff simulation, especially the runoff peak which is the major concern of real-time flood forecasting. Moreover, by explicitly take consideration of different atmospheric and underlying surface conditions, XAJ-EB is more suitable than XAJ for the study of hydrological responses under changing climate/land cover, which  extend the applications of the original XAJ model.

*Author contributions.* Zhang and Corbari designed and supervised the study; Fang, Cobari, Mancini and Niu developed the model and conducted simulations; Zeng collected and processed the remote sensing and ground measured data; all authors contributed to the preparation of this manuscript.

*Acknowledgements.* This study is funded by the Major International (Regional) Joint Research Project of the National Natural Science Foundation of China (51420105014), the Special Scientific Research Fund of Ministry of Water Resources' Public Welfare Profession of China (201401034), ESA-MOST Dragon 3 programme (10664), the National Natural Science Foundation of China (51609175), the Open Foundation of State Key Laboratory of Hydrology-Water Resources and Hydraulic Engineering (2015490211). We also appreciate two referees for their valuable comments and suggestions.

[revised manuscript text omitted]